# Arabidopsis heterotrimeric G proteins regulate immunity by directly coupling to the FLS2 receptor

Xiangxiu Liang[1†], Pingtao Ding[2†], Kehui Lian[2], Jinlong Wang[1], Miaomiao Ma[1], Lin Li[3], Lei Li[1], Meng Li[1], Xiaojuan Zhang[1], She Chen[3], Yuelin Zhang[2*], Jian-Min Zhou[1*]

[1]State Key Laboratory of Plant Genomics, Institute of Genetics and Developmental Biology, Chinese Academy of Sciences, Beijing, China; [2]Department of Botany, University of British Columbia, Vancouver, Canada; [3]National Institute of Biological Sciences, Beijing, China

**Abstract** The Arabidopsis immune receptor FLS2 perceives bacterial flagellin epitope flg22 to activate defenses through the central cytoplasmic kinase BIK1. The heterotrimeric G proteins composed of the non-canonical Gα protein XLG2, the Gβ protein AGB1, and the Gγ proteins AGG1 and AGG2 are required for FLS2-mediated immune responses through an unknown mechanism. Here we show that in the pre-activation state, XLG2 directly interacts with FLS2 and BIK1, and it functions together with AGB1 and AGG1/2 to attenuate proteasome-mediated degradation of BIK1, allowing optimum immune activation. Following the activation by flg22, XLG2 dissociates from AGB1 and is phosphorylated by BIK1 in the N terminus. The phosphorylated XLG2 enhances the production of reactive oxygen species (ROS) likely by modulating the NADPH oxidase RbohD. The study demonstrates that the G proteins are directly coupled to the FLS2 receptor complex and regulate immune signaling through both pre-activation and post-activation mechanisms.

**\*For correspondence:** yuelin. zhang@botany.ubc.ca (YZ); jmzhou@genetics.ac.cn (JMZ)

[†]These authors contributed equally to this work

**Competing interests:** The authors declare that no competing interests exist.

## Introduction

As an intensely studied Pattern Recognition Receptor (PRR) in plants, FLS2 serves as an excellent model understanding plant innate immune signaling and receptor kinases in general (*Macho and Zipfel, 2014*). It forms a dynamic complex with the co-receptor BAK1 and the receptor-like cytoplasmic kinase BIK1 to perceive a conserved bacterial flagellar epitope, flg22, to activate a variety of defense responses (*Chinchilla et al., 2007*; *Heese et al., 2007*; *Lu et al., 2010*; *Zhang et al., 2010*; *Sun et al., 2013*). Stability of FLS2 and BIK1 is subject to regulation by ubiquitin-proteasome system and a calcium-dependent protein kinase (*Lu et al., 2011*; *Monaghan et al., 2014*). We and others previously showed that BIK1 directly phosphorylates the NADPH oxidase RbohD to prime flg22-induced reactive oxygen species (ROS; *Kadota et al., 2014*; *Li et al., 2014*).

Heterotrimeric G proteins are central for signaling in animals (*McCudden et al., 2005*; *Oldham et al., 2008*), which contain hundreds of G Protein-Coupled Receptors (GPCRs). In the pre-activation state, the GDP-bound Gα interacts with the Gβγ dimer to form a heterotrimer. Upon activation by GPCR, Gα exchanges GDP for GTP, resulting in the activation of the heterotrimer. The activated Gα and Gβγ dissociate from each other to regulate downstream effectors. Plants contain canonical Gα (encoded by *GPA1* in Arabidopsis), Gβ (encoded by *AGB1* in Arabidopsis), Gγ proteins (encoded by *AGG1*, *AGG2*), and a non-canonical Gγ (encoded by *AGG3* in Arabidopsis) (Urano and Jones, 2013). Plants additionally encode extra-large G proteins (XLGs, encoded by *XLG1*, *XLG2*, and *XLG3* in Arabidopsis) that carry a variable N-terminal domain and a C-terminal Gα domain (*Lee and*

**eLife digest** Living cells need to be able to detect changes in their environment and respond accordingly. This ability involves signals from outside of the cell triggering changes to the activity inside the cell. Heterotrimeric G proteins are important for this kind of signaling in a wide range of organisms. In animals and fungi, these proteins directly work with a specific class of receptor proteins called G protein-coupled receptors (or GPCRs for short). Plants also have heterotrimeric G proteins, but it remains unclear whether they similiarly work with GPCRs.

Plants detect invading microbes by using receptors that are completely different from GPCRs. For example, a receptor called FLS2 from the model plant Arabidopsis senses a telltale protein produced by bacteria, and then passes the signal to another protein called BIK1 to activate the plant's defenses. Heterotrimeric G proteins are required for this process, but the underlying mechanisms remain unknown.

Liang, Ding et al. now show that heterotrimeric G proteins regulate FLS2-controlled defenses by directly interacting with FLS2 and BIK1. Heterotrimeric G proteins also enhance defenses in at least two different ways. Firstly, in the absence of an infection, heterotrimeric G proteins stabilize the BIK1 protein to ensure that it is ready to respond. Secondly, if FLS2 does detect the telltale bacterial protein, BIK1 marks one of the heterotrimeric G proteins with a phosphate group. This then allows the G protein to boost the activity of another plant enzyme that is vital for defense signaling.

In the future, it will be important to work out how activation of FLS2 leads to the activation of heterotrimeric G proteins. Furthermore, heterotrimeric G proteins are likely to regulate additional plant proteins when defenses are activated, and further studies are needed to identify these proteins.

*Assman, 1999*; *Ding et al., 2008*). Recent advances indicate that the Arabidopsis XLGs are functional Gα proteins and interact with Gβγ dimers to form heterotrimers (*Zhu et al., 2009*; *Maruta et al., 2015*; *Chakravorty et al., 2015*). Heterotrimeric G proteins play important roles in a variety of biological processes in plants, including cell division (*Ullah et al., 2001*, *2003*; *Chen et al., 2003*), meristem maintenance (*Bommert et al., 2013*), root morphogenesis (*Ding et al., 2008*), seed development and germination (*Chen et al., 2006*; *Pandey et al., 2006*), nitrogen assimilation (*Sun et al., 2014*), and response to ABA (*Wang et al., 2001*), low temperature (*Ma et al., 2015*), and blue light (*Warpeha et al., 2006*).

Accumulating evidence indicate that heterotrimeric G proteins also play an important role in plant disease resistance against diverse pathogens (*Llorente et al., 2005*; *Trusov et al., 2006*; *Zhu et al, 2009*; *Ishikawa, 2009*; *Cheng et al., 2015*). Recent reports indicate that XLG2, AGB1, and AGG1/2 mediate immune responses downstream of PRRs (*Ishikawa, 2009*; *Zhu et al, 2009*; *Liu et al., 2013*; *Lorek et al., 2013*; *Torres et al., 2013*; *Maruta et al., 2015*). XLG2 was first shown to play an important role in basal resistance to *P. syringae* (*Zhu et al., 2009*). A recent report showed that *XLG2*, but not *XLG1*, is required for resistance to *P. syringae* and flg22-induced ROS production (*Maruta et al., 2015*). *AGB1* and *AGG1/2,* but not *AGG3*, are required for resistance to *P. syringae* and microbial pattern-induced ROS production (*Liu et al., 2013*; *Lorek et al., 2013*; *Torres et al., 2013*). Furthermore, epistatic analyses indicated that *AGB1* acts in the same pathway as RbohD (*Torres et al., 2013*). However, it is still debated whether plants possess 7 transmembrane GPCRs (*Taddese et al., 2014*; *Urano and Jones, 2014*). One recent report suggests that GPA1, AGG1/2 can interact with BAK1 and the chitin-binding receptor kinase CERK1, but not FLS2 (*Aranda-Sicilia et al., 2015*). However, GPA1 does not appear to play a role in flg22-induced ROS and disease resistance to *P. syringae* (*Liu et al., 2013*; *Torres et al., 2013*). How XLG2 and AGB1 regulate PRR-mediated immunity remains elusive.

In this study, we report XLG2, AGB1, and AGG1/2 modulates flg22-triggered immunity by directly coupling to the FLS2-BIK1 receptor complex. Prior to activation by flg22, the G proteins attenuate the proteasome-dependent degradation of BIK1, ensuring optimum signaling competence. After flg22 stimulation, XLG2 dissociates from AGB1, indicating a ligand-induced dissociation of Gα from Gβγ. In addition, we provide evidence that activation by flg22 additionally leads to XLG2

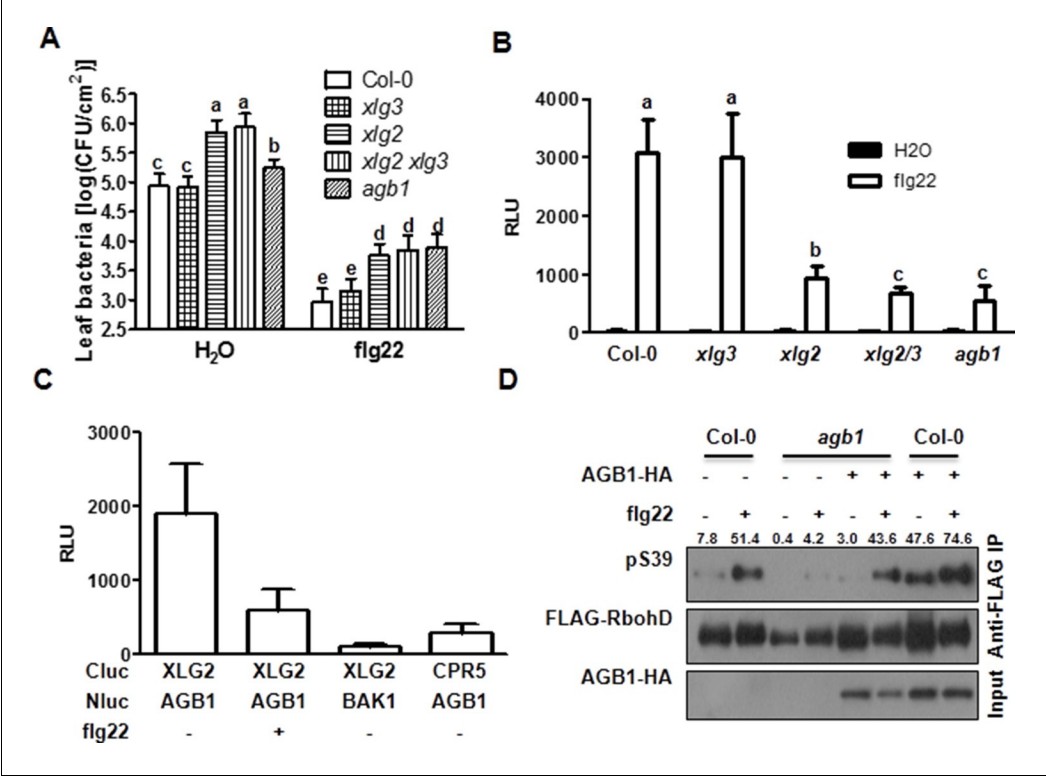

**Figure 1.** G proteins are required for FLS2-mediated immunity. (**A**) *XLG2/3* and *AGB1* play overlapping but not identical roles in disease resistance to *Pst*. Plants of indicated genotypes were infiltrated with $H_2O$ and flg22 1 day before infiltration with *P. syringae* DC3000, and bacteria number was determined 2 days later (mean ± SD; n ≥ 6; p<0.05, Student's t-test; different letters indicate significant difference). (**B**) *xlg2/3* and *agb1* are similarly compromised in flg22-induced ROS burst. Leaves of the indicated genotypes were examined for flg22-induced ROS production, and peak RLU values are shown (mean ± SD; n ≥ 6; p<0.05, Student's t-test; different letters indicate significant difference). (**C**) Flg22 treatment disrupts XLG2-AGB1 interaction. Cluc-XLG2 and AGB1-HA-Nluc constructs are transiently expressed in *Nb* leaves, relative luminescence unit (RLU) was measured 2 days later. Cluc-CPR5 and BAK1-HA-Nluc were used as negative control (mean ± SD; n ≥ 6). (**D**) Flg22-induced RbohD phosphorylation is impaired in *agb1*. FLAG-RbohD and/or AGB1-HA constructs were expressed under control of the 35S promoter in WT or *agb1* protoplasts. The FLAG-RbohD protein was affinity purified and subject to anti-FLAG and anti-pSer39 immuoblot analyses. Numbers indicate arbitrary units of RbohD pS39 phosphorylation calculated from densitometry measurements normalized to total FLAG-RbohD protein. Each experiment was repeated three times, and data of one representative experiment are shown.

The following source data and figure supplements are available for figure 1:

**Source data 1.** Raw data and exact p value of *Figure 1A*, *B* and *Figure 1—figure supplement 1*.

**Figure supplement 1.** Flg22-induced ROS burst is compromised in *xlg2* plants.

**Figure supplement 2.** *XLG2/3* and *AGB1*, but not *XLG1*, are transcriptionally induced by flg22.

**Figure supplement 3.** XLG2/3 interact with AGB1 through both N and C termini.

phosphohrylation by BIK1, and this phosphorylation positively regulates RbohD-dependent ROS production. Together the study illustrates two distinct mechanisms underlying the G protein-mediated regulation of the FLS2 signaling.

## Results

### Characterization of XLG2 and AGB1 in FLS2-mediated immunity

To identify additional components of the FLS2 immune pathway, we conducted a reverse genetic screen for mutants that were compromised in flg22-induced disease resistance to *Pseudomonas syringae* pv *tomato* (*Pst*). One mutant displayed significantly reduced resistance was *xlg2* (*Figure 1A*), confirming previous report by *Zhu et al. (2009)*. Further characterization indicated that the mutant is compromised in flg22-induced ROS (*Figure 1B*, *Figure 1—figure supplement 1*), confirming results reported previously (*Maruta et al., 2015*). An examination of gene expression showed that *XLG2, XLG3,* and *AGB1,* but not *XLG1,* were induced in response to flg22 treatment (*Figure 1—figure supplement 2*). This is in agreement with the recent report that *XLG2/3,* but not *XLG1,* are required for flg22-induced responses and disease resistance to *P. syringae* (*Maruta et al., 2015*). A comparison of *xlg2* and *agb1* mutant showed that the two mutants were similarly compromised in flg22-induced ROS burst and resistance against *Pst,* supporting that they act together to regulate FLS2 immunity. Interestingly, *xlg2* plants were more susceptible to *Pst* than *agb1* plants in the absence of flg22 treatment (*Figure 1A*), suggesting that XLG2 plays additional role in plant immunity independent of AGB1. As reported previously (*Maruta et al., 2015*), we found that the *xlg3* mutant was similar to WT in *Pst* resistance and flg22-induced ROS, whereas the *xlg2 xlg3* double mutant was slightly more defective in *Pst* resistance (*Figure 1A*) and flg22-induced ROS (*Figure 1B*), indicating that *XLG2/3* play additive roles in flg22-induced immunity. Because *AGG1/2,* but not *AGG3* and *GPA1,* are required for flg22-induced defense responses and *Pst* resistance (*Liu et al., 2013*; *Torres et al., 2013*), these results confirmed that the heterotrimeric G proteins required for FLS2 signaling and *Pst* resistance include the non-canonical Gα proteins XLG2/3, Gβ protein AGB1, and Gγ proteins AGG1/2.

Previous yeast two-hybrid, yeast tri-hybrid and Bi-Fluorescence Complementation assays showed that XLG2/3 form heterotrimers with the Gβγ dimer through AGB1 (*Zhu et al., 2009*; *Maruta et al., 2015*; *Chakravorty et al., 2015*). Consistent with this, luciferase complementation and co-immuno-precipitation assays detected XLG2-AGB1 and XLG3-AGB1 interactions in *Nicotiana Benthamiana* (*Nb*) plants (*Figure 1C*, *Figure 1—figure supplement 3A*) and Arabidopsis protoplasts (*Figure 1—figure supplement 3B–C*). Both the N and C termini of XLG2 are sufficient for the interaction with AGB1 (*Figure 1—figure supplement 3C*). Note that XLG2-FLAG and AGB1-HA used in co-IP assays retain their biological functions when introduced into *agb1* protoplasts or *xlg2* plants (see below in *Figures 1D* and *5E*). In the animal model, the heterotrimeric G protein is dynamically regulated upon ligand stimulation of GPCR. We tested whether flg22 treatment can similarly regulate XLG2-AGB1 interaction. As shown in *Figure 1C*, the flg22 treatment resulted in a great reduction in XLG2-AGB1 interaction in *Nb* plants, suggesting that Arabidopsis heterotrimeric G proteins are reminiscent of their animal counterparts in that the heterotrimer is dynamically regulated upon ligand activation of receptors. Because the heterotrimeric G protein mutants under investigation all display defects in flg22-induced ROS, which requires phosphorylation of the NADPH oxidase RbohD by the BIK1 kinase (*Kadota et al., 2014*; *Li et al., 2014*), we monitored the flg22-induced phosphorylation of RbohD in *agb1* using antibodies that recognize the BIK1-specific S39 phosphorylation. Flg22-induced S39 phosphorylation was significantly reduced in *agb1* mutant protoplasts, whereas complementation of the mutant protoplasts with *AGB1* restored RbohD S39 phosphorylation (*Figure 1D*). Consistent with a positive role of AGB1 in flg22-induced RbohD S39 phosphorylation, transient over-expression of *AGB1* in WT protoplasts resulted in constitutive S39 phosphorylation in RbohD (*Figure 1D*). Together these results suggest that the hetero-trimeric G proteins regulate early components of the FLS2 signaling pathway.

### Dynamic and direct interaction between XLG2 and the FLS2-BIK1 complex

To identify potential proteins regulated by XLG2, we performed immunoprecipitation of XLG2-FLAG transiently expressed in protoplasts. LC-MS/MS analysis of the immune complex identified four peptides corresponding to PBL20 (*Figure 2—figure supplement 1A*), a homolog of BIK1. In contrast, immunoprecipitation of the control protein CPR5-FLAG failed to identify any PBL20 peptides. Co-IP and luciferase complementation assays confirmed that XLG2/3 indeed interacted strongly with

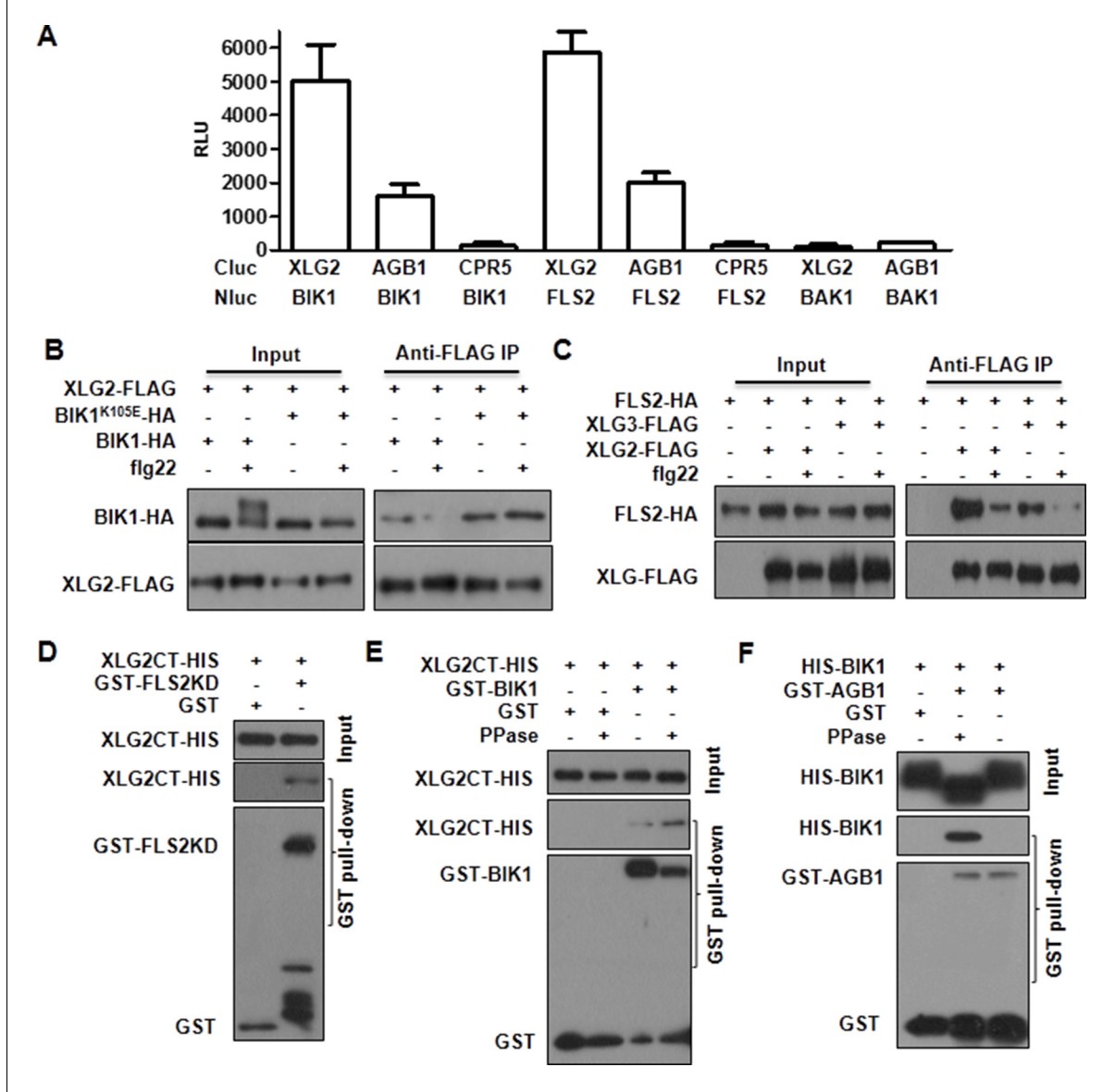

**Figure 2.** Flg22 regulates interactions between G proteins and the FLS2-BIK1 receptor complex. (**A**) XLG2 and AGB1 interact with BIK1 and FLS2 in *Nb* plants. The indicated Nluc and Cluc constructs were transiently expressed in *Nb* plants for luciferase complementation assay. Relative luminescence unit (RLU) shows the strength of protein-protein interaction (mean ± SD; n ≥ 6. (**B**) XLG2 interacts with BIK1 in Arabidopsis protoplasts and the interaction is dynamically regulated by flg22. (**C**) XLG2/3 interact with FLS2 in Arabidopsis protoplasts and the interaction is dynamically regulated by flg22. The indicated constructs were co-expressed in WT protoplasts, and Co-IP assays were performed using agarose-conjugated anti-FLAG antibody. BIK1$^{K105E}$ carries a mutation in the ATP-binding site. (**D**) The C terminus of XLG2 directly interacts with FLS2 kinase domain. XLG2CT-HIS (amino acids 459–861) was incubated with GST or GST-FLS2KD (FLS2 kinase domain) for GST pull-down assay and detected by anti-HIS and anti-GST immunoblots. (**E**) XLG2 primarily interacts with non-phosphorylated BIK1. XLG2CT-HIS was incubated with GST or GST-BIK1 that was untreated or pre-treated with λ phosphatase (PPase), and GST pull-down assay was performed. (**F**) AGB1 interacts with the non-phosphorylated BIK1. Untreated or PPase-treated BIK1-HIS was incubated with GST or GST-AGB1, and GST pull-down assay was performed. Each experiment was repeated two (**D–F**) or three (**A–C**) times, and data of one representative experiment are shown.

The following figure supplements are available for figure 2:

**Figure supplement 1.** PBL20 interacts with G proteins.

**Figure supplement 2.** XLG3 interacts with BIK1.

**Figure supplement 3.** XLG2 interacts with FLS2 and BIK1 primarily through the C terminus.

PBL20 (*Figure 2—figure supplement 1B–C*). AGB1 also interacted with PBL20, albeit much weaker than did XLG2. Because PBL20 and BIK1 belong to the same family of receptor like cytoplasmic kinases (RLCKs) and share similar biochemical properties (*Zhang et al., 2010*), we therefore tested whether BIK1 also interacts with the G proteins. Luciferase complementation assays showed that XLG2/3 strongly interacted with BIK1 but not BAK1 or CPR5 in *Nb* plants (*Figure 2A*, *Figure 2—figure supplement 2A*), whereas AGB1 interacted with BIK1 at a much lower level. Similarly, FLS2 also displayed strong interactions with XLG2/3 and a weaker interaction with AGB1 (*Figure 2A*, *Figure 2—figure supplement 2A*). The interactions were further confirmed by Co-IP assays (*Figure 2B–C*, *Figure 2—figure supplement 2B*). In the absence of flg22 treatment, XLG2/3 strongly interacted with BIK1 and FLS2 (*Figure 2B–C*, *Figure 2—figure supplement 2B*). Treatment of protoplasts with flg22 led to a much weaker interaction, indicating that flg22 induces dissociation of XLG2 from the FLS2 receptor complex. The BIK1$^{K105E}$-HA mutant protein, which is defective in ATP-binding and dominantly inhibits flg22-induced signaling when expressed in protoplasts (*Zhang et al., 2010*), interacted with XLG2 and XLG3 irrespective of the presence or absence of flg22, suggesting that the activation of the BIK1 kinase is required for the dissociation.

We next tested XLG2 domains for interactions with BIK1 and FLS2. Luciferase complementation and Co-IP assays showed that the C terminus of XLG2 interacted strongly with BIK1 and FLS2, whereas the N terminus of XLG2 displayed weak interactions (*Figure 2—figure supplement 3A–C*), indicating that XLG2 interacts with FLS2 and BIK1 primarily through its C terminus. Similar to the full-length XLG2 protein, the C terminus of XLG2 also dissociated from BIK1 when protoplasts were treated with flg22 (*Figure 2—figure supplement 3C*). GST pull-down assays showed that the HIS-tagged C-terminal fragment of XLG2 strongly interacted with GST-tagged FLS2 kinase domain, but not GST alone (*Figure 2D*), indicating that the C terminus of XLG2 directly interacts with FLS2. Surprisingly, GST pull-down assay showed a weak interaction between XLG2 and BIK1 (*Figure 2E*). Interestingly, phosphatase treatment of BIK1 enhanced its interaction with XLG2 (*Figure 2E*), suggesting that XLG2 has greater affinity with non-phosphorylated BIK1. We similarly tested interaction between AGB1 with untreated and de-phosphorylated BIK1 by GST pull-down assay. While the untreated BIK1 is unable to interact with AGB1, the de-phosphorylated form of BIK1 strongly interacted with AGB1 (*Figure 2F*). Together the results show that the G proteins dynamically interact with the FLS2-BIK1 receptor complex during FLS2 signaling, and the findings are consistent with the observed defects in flg22-induced responses and RbohD phosphorylation in the G protein mutants.

## XLG2/3, AGB1, and AGG1/2 maintain BIK1 stability by attenuating the proteasome-dependent degradation of BIK1

The direct interaction of XLG2/3 with the FLS2 receptor complex prompted us to examine FLS2, BIK1 and BAK1 proteins in the G protein mutants. To examine BIK1 accumulation, we crossed *agb1* with a transgenic line carrying a *BIK1-HA* transgene under control of the native *BIK1* promoter (*Zhang et al., 2010*) and identified sibling transgenic lines of either *agb1* or *AGB1* genotype. Immunoblot analyses showed that the BIK1-HA protein accumulated to a much lower level in *agb1* compared to WT (*Figure 3A*), indicating that *AGB1* is required for BIK1 accumulation in plants. In contrast, FLS2 and BAK1 protein accumulation was not affected in *agb1,* indicating a specific effect on BIK1 protein. Examination of *BIK1-HA* transcripts indicated that the transgene is similarly expressed in *agb1* and WT plants (*Figure 3—figure supplement 1A*). To further rule out the possibility that AGB1 modulates BIK1 at transcriptional level, we transformed *agb1* and WT plants with *BIK1-HA* transgene under the control of the 35S promoter. Again, a much lower BIK1-HA accumulation was observed in *agb1* than in WT plants although the *BIK1-HA* transcripts accumulated to similar levels in the two lines (*Figure 3—figure supplement 1B*). Similarly, the BIK1-HA protein accumulation was greatly reduced in *agg1 agg2* double mutant plants (*Figure 3B*), indicating that *AGG1/2* are also required for BIK1 accumulation. We further asked whether *XLG2* and *XLG3* are required for BIK1 accumulation by transiently expressing BIK1-HA in *xlg2 xlg3* mutant protoplasts. Regardless of the promoter used for BIK1-HA expression, BIK1 accumulated to lower levels in the *xlg2 xlg3* double mutant than in WT protoplasts (*Figure 3C*), indicating that *XLG2/3* are also required for BIK1 accumulation. The defect in protein accumulation is specific to BIK1, as BAK1 accumulation was not affected in *xlg2 xlg3* protoplasts (*Figure 3C*). The accumulation of BIK1-HA was not affected in *gpa1* mutant plants (*Figure 3—figure supplement 1C*), a result that is consistent with previous report that GPA1 is not required for flg22-induced ROS (*Liu et al., 2013*;

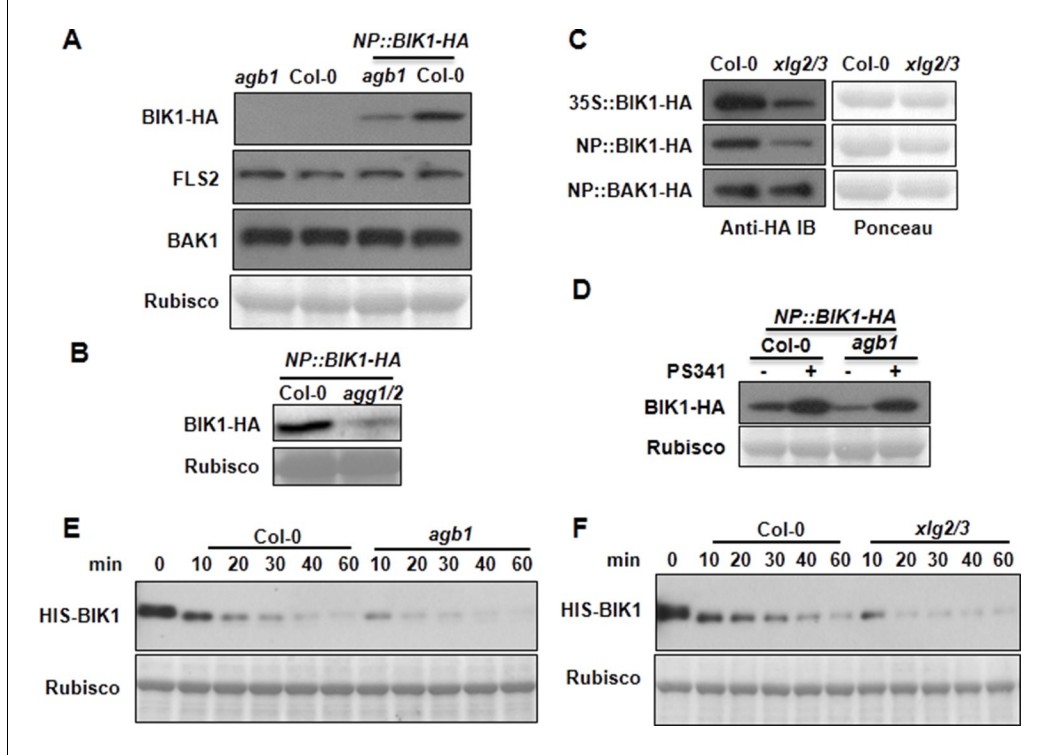

**Figure 3.** G proteins positively regulate immunity and BIK1 stability. (A) *AGB1* is required for accumulation of BIK1, but not FLS2 and BAK1. *BIK1-HA* was introduced into *agb1* by crossing, homozygotes of the indicated genotypes in F3 generation were used for immunoblot analyses. (B) *AGG1/2* are required for BIK1 stability. *NP::BIK1-HA* was introduced into *agg1/2* by crossing, homozygous plants in F3 generation were subject to immunoblot analyses. (C) *XLG2/3* are required for BIK1 accumulation. NP::BIK1-HA, 35S::BIK1-HA and NP::BAK1-HA plasmids were transiently expressed in WT and *xlg2/3* protoplasts, and accumulation of BIK1 and BAK1 was determined by immunoblot analyses. (D) AGB1 regulates BIK1 accumulation through the proteasome pathway. One-week-old *NP::BIK1-HA* seedlings of WT (Col-0) or *agb1* background were pretreated with DMSO (-) or 100 µM proteasome inhibitor PS341(+) for 8 hr before total protein was isolated for immunoblot analysis. (E) The *agb1* extract shows accelerated degradation of BIK1 in vitro (F) The *xlg2 xlg3* extract shows accelerated degradation of BIK1. Total extracts from WT (Col-0), *agb1* and *xlg2 xlg3* seedlings were incubated with HIS-BIK1 protein at 22°C for the indicated times, and equal amounts of sample were analyzed using anti-HIS immunoblot. Each experiment was repeated at least three times, and data from one representative experiment are shown.

The following figure supplements are available for figure 3:

**Figure supplement 1.** G proteins are required for BIK1 stability.

**Figure supplement 2.** AGB1 regulates BIK1 stability through proteasome pathway.

**Figure supplement 3.** *AGB1* and *XLG2/3* attenuate PBL20 degradation.

*Torres et al., 2013*). BIK1 is known to be turned-over through the ubiquitin-proteasome pathway (*Monaghan et al., 2014*). Treatment of plants with proteasome inhibitors PS341 (*Figure 3D*) and MG132 (*Figure 3—figure supplement 2A*) restored BIK1 accumulation in *agb1* mutant plants, suggesting that AGB1 regulates BIK1 stability through the proteasome pathway. We further performed *in vitro* protein degradation assay to compare the rate of BIK1 degradation in total protein extracts from WT and mutant plants. The recombinant BIK1 was degraded in the WT extract, and this degradation was blocked by PS341 (*Figure 3—figure supplement 2B*), recapitulating the proteasome-dependent degradation of BIK1 in plants. The rate of BIK1 degradation was more rapid in *agb1* and *xlg2 xlg3* extracts compared to that in the WT extract (*Figure 3E–F*). Because XLG2 also interacted with PBL20, we further tested whether XLG2/3 and AGB1 similarly regulate PBL20 stability. Indeed, PBL20 was also degraded in WT extracts, and this degradation was enhanced when extracts from *xlg2 xlg3* or *agb1* mutants were used (*Figure 3—figure supplement 3A–B*). Together these results

demonstrate that the heterotrimeric G proteins formed by XLG2/3, AGB1, and AGG1/2 attenuate proteasome-dependent degradation of BIK1 and PBL20.

## BIK1 accumulation accounts for the G protein-mediated regulation of immunity

We next tested whether the regulation of BIK1 stability contributes to the role of heterotrimeric G proteins in flg22-induced immune responses. Transient overexpression of BIK1 in *xlg2 xlg3* and *agb1* protoplasts largely restored the flg22-induced RbohD S39 phosphorylation (*Figure 4A*), indicating that the heterotrimeric G proteins positively regulate RbohD phosphorylation at least in part by enhancing BIK1 stability. We next asked whether increased copy number of *BIK1* could rescue flg22-induced ROS burst in *agb1* mutant. Introgression of the *NP::BIK1-HA* transgene into *agb1* by crossing significantly enhanced flg22-induced ROS production in these plants compared to *agb1* plants that contained only the endogenous copy of *BIK1* (*Figure 4B*), indicating that increasing BIK1 accumulation by the transgene at least partially restored flg22-induced ROS in *agb1*. Col-0 *NP::BIK1-HA* plants displayed higher level of ROS compared to non-transgenic WT plants (*Figure 4B*), indicating that BIK1 protein level is rate-limiting in flg22-induced immune responses. We further tested the impact of the *NP::BIK1-HA* transgene on flg22-induced resistance to *Pst*. While the *agb1* consistently supported greater amount of bacterial growth compared to WT plants following flg22 treatment, the *agb1 NP::BIK1-HA* plants were indistinguishable from WT (*Figure 4C*). Col-0 *NP::BIK1-HA* plants supported less bacterial growth than did WT plants, further supporting the importance of BIK1 abundance in FLS2-mediated immunity. We similarly introduced the *NP::BIK1-HA* transgene into *xlg2 xlg3* double mutant plants by transformation. The resulting transgenic lines displayed near WT ROS production (*Figure 4D*) and *Pst* resistance (*Figure 4E*) in response to flg22 treatment, indicating that *XLG2/3* also regulate FLS2-mediated immunity by controlling BIK1 stability.

## XLG2 is phosphorylated in the N terminus upon flg22 treatment

BIK1 is postulated to phosphorylate multiple proteins, including RbohD, in PTI (Pattern-Triggered Immunity) signaling (*Kadota et al., 2014*; *Li et al., 2014*; *Macho and Zipfel, 2014*). An examination of XLG2 N-terminal fragment expressed in protoplasts following flg22 treatment showed a prominent shift in protein migration in SDS-PAGE (*Figure 5A*). The shift of protein mobility was sensitive to phosphatase treatment, which is indicative of flg22-induced phosphorylation in XLG2. The flg22-induced phosphorylation similarly occurred to the N terminus of XLG3, but not XLG1 (*Figure 5—figure supplement 1*), which is consistent with the notion that XLG2/3 but not XLG1, are involved in flg22 signaling. To determine phospho-sites in XLG2, we expressed full-length XLG2-FLAG in protoplasts. Following flg22 treatment, the XLG2-FLAG protein was affinity-purified and subject to LC-MS/MS analysis for phospho-peptides. We were able to recover 87% of total peptide sequence, among which we identified 23 phospho-peptides corresponding to at least 15 phospho-sites (*Figure 5—figure supplement 2*). Interestingly, all but two phospho-peptides were located in the N terminus. Strikingly, phospho-sites between amino acids 141–156 accounted for nearly half of the phospho-peptides. We further tested whether these are the major phospho-sites by site-directed mutagenesis. Simultaneously substituting three to four of these residues into non-phosphorylatable alanine (XLG2$^{S148A,S150A,S151A}$; XLG2$^{S141A,S148A,S150A,S151A}$) resulted in severe reduction to complete elimination in flg22-induced band shift in XLG2 N terminus (*Figure 5B*), indicating that amino acids Ser141, Ser148, Ser150 and Ser151 are indeed required for overall phosphorylation of XLG2 in response to flg22 treatment.

We next tested whether BIK1 is able to phosphorylate XLG2 N terminus in vitro Radio-labelling assay showed that the recombinant BIK1, but not BIK1$^{K105E}$, strongly phosphorylated the N terminus of XLG2 (*Figure 5C*). To determine whether the major phospho-sites in XLG2 were also phosphorylated by BIK1, we raised antibodies that specifically recognize phospho-S148 and phospho-S150. Immunoblot analysis showed that S148 and S150 were indeed phosphorylated in vitro by BIK1, but not BIK1$^{K105E}$ (*Figure 5D*).

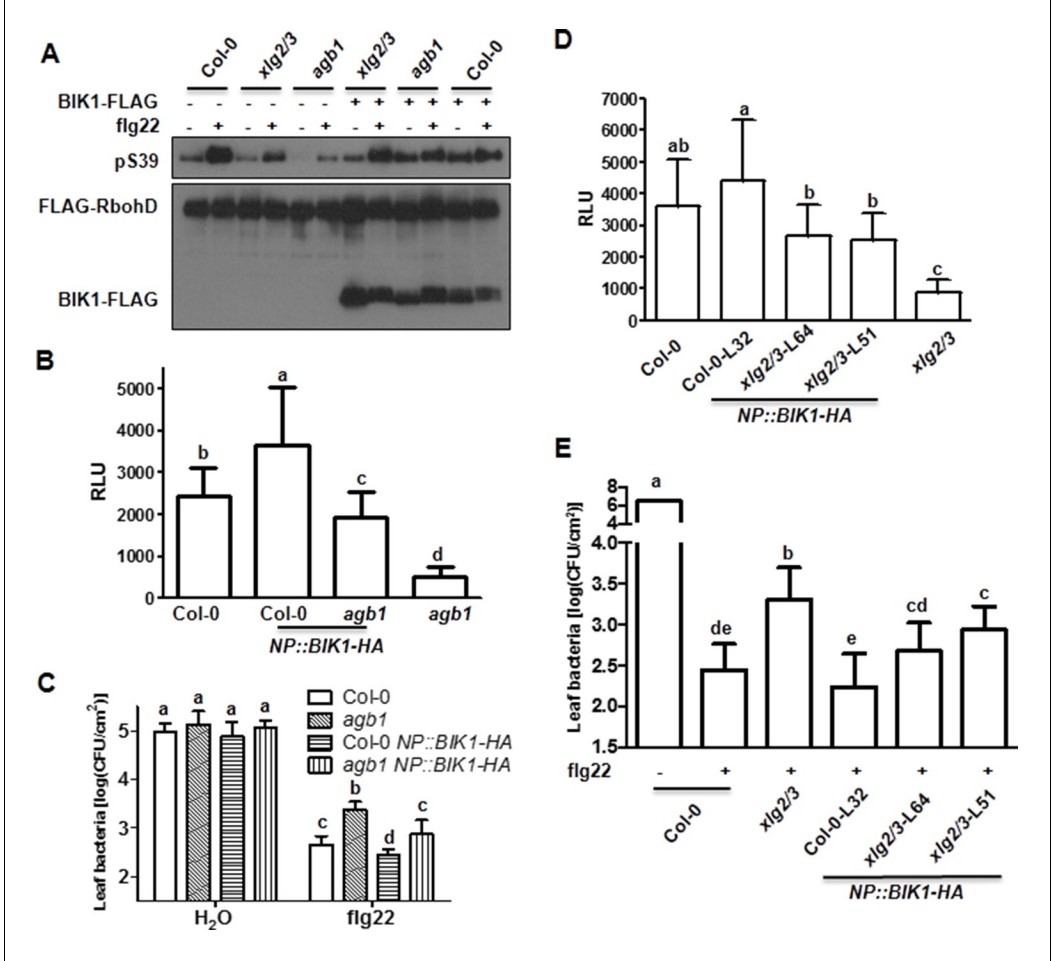

**Figure 4.** BIK1 level accounts for G protein-mediated regulation of FLS2 immunity. (**A**) Transient expression of BIK1 in *agb1* and *xlg2/3* mutant protoplasts restores RbohD phosphorylation. FLAG-RbohD and BIK1-HA constructs are transiently expressed in protoplasts from WT (Col-0), *agb1* and *xlg2/3*. FLAG-RbohD protein was affinity purified and detected by anti-FLAG and anti-pSer39 immuoblots. (**B**) *BIK1* transgene restores flg22-induced ROS burst in *agb1*. (**C**) *BIK1* transgene partially restores flg22-induced resistance to *Pst* in *agb1*. *NP::BIK1-HA* was introduced into *agb1* by crossing, transgenic lines of *agb1* or Col-0 background in the F3 generation were used for the assays. (**D**) *BIK1* transgene partially restores flg22-induced ROS burst in *xlg2 xlg3* mutant. (**E**) *BIK1* transgene partially restores flg22-induced resistance to *Pst*. The *NP::BIK1-HA* transgene was introduced into WT (Col-0-L32) and *xlg2 xlg3* (*xlg2/3*-L64 and *xlg2/3*-L51) plants by *Agrobacterium*-mediated transformation. Independent T2 transgenic lines were used for the assays. Peak relative luminescence unit (RLU) values were shown for ROS assays (**B** and **D**) and leaf bacterial populations 2 days after bacterial inoculation were shown for flg22-protection assays (**C** and **E**). Bars in B-E represent mean ± SD (n ≥ 6; p<0.05, Student's t-test; different letters indicate significant difference). Each experiment was repeated two (**A**) or three (**B–E**) times, and data of one representative experiment are shown.

The following source data is available for figure 4:

**Source data 1.** Raw data and exact p value of *Figure 4B—E*.

## XLG2 phosphorylation is required for disease resistance to *Pst* and optimum ROS production upon flg22 induction

We introduced WT, non-phosphorylatable (XLG2$^{S141A,S148A,S150A,S151A}$) and phospho-mimicking (XLG2$^{S141D,S148D,S150D,S151D}$) forms of *XLG2-FLAG* transgene under control of the native *XLG2* promoter into *xlg2* plants and analyzed flg22-induced ROS production in the resulting transgenic lines. While the *xlg2* transgenic lines carrying the WT or phospho-mimicking *XLG2-FLAG* transgene completely restored the flg22-induced ROS to WT level, the two lines transformed with the non-phosphorylatable *XLG2-FLAG* transgene only partially restored the flg22-induced ROS (*Figure 5E*). The inability of phospho-mimicking XLG2 to constitutively activate ROS suggests that the phosphorylation of XLG2 is required, but not sufficient, for optimum activation of RbohD. We further

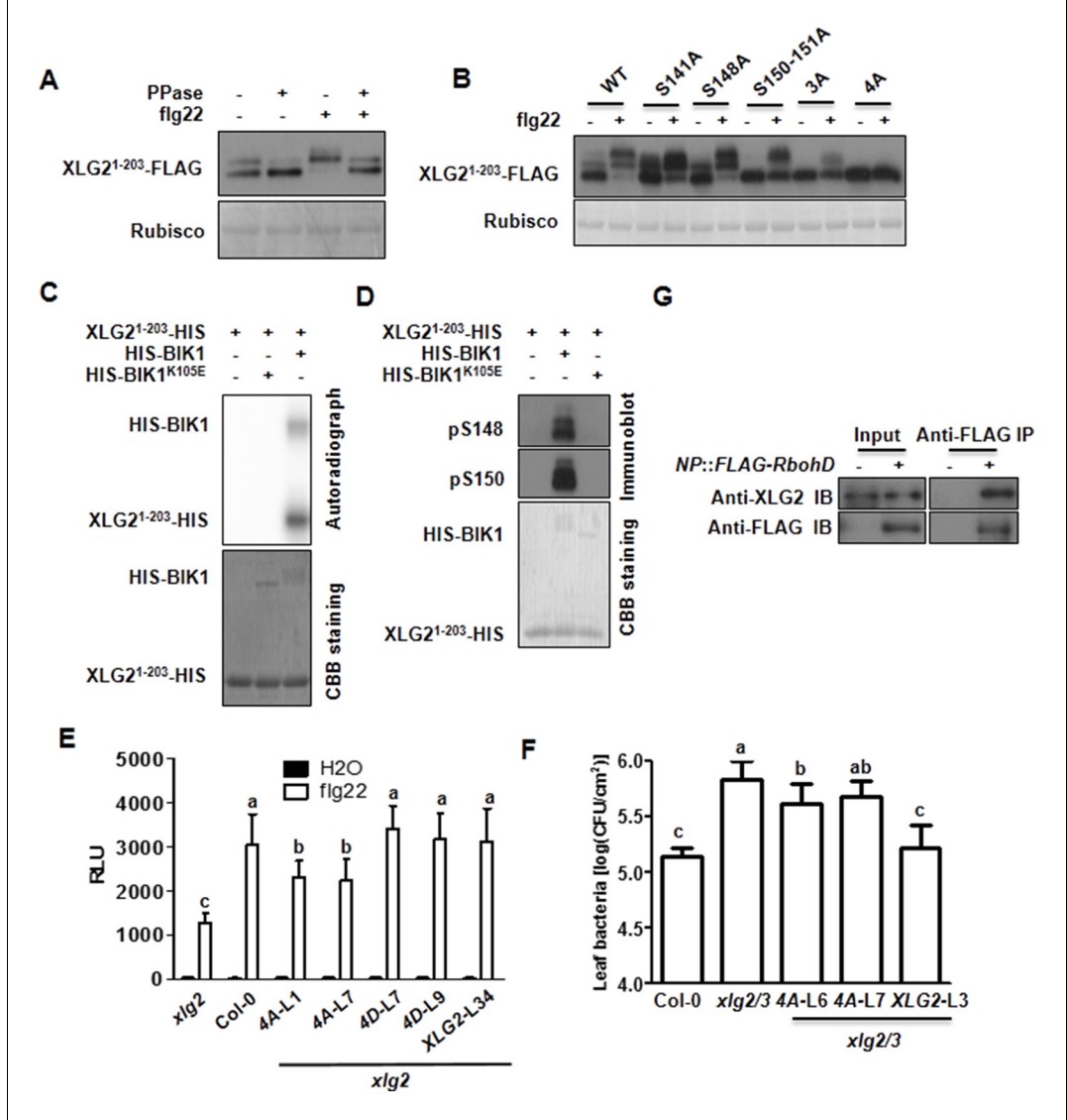

**Figure 5.** Phosphorylation of XLG2 by BIK1 regulates flg22-induced ROS. (A) Flg22-induces phosphorylation of XLG2 in the N terminus. Protoplasts expressing XLG2$^{1-203}$-FLAG were treated with flg22. The total protein was treated with (+) or without (-) λ protein phosphatase (PPase) prior to anti-FLAG immunoblot analysis. (B) Flg22-induced phosphorylation of XLG2 in protoplasts primarily occurs in Ser141, Ser148, Ser150 and Ser151. Different mutated form of XLG2$^{1-203}$-FLAG constructs were transiently expressed in WT protoplast, treated with flg22 and the migration of XLG2$^{1-203}$-FLAG were examined by anti-FLAG immunoblot. (C) BIK1 phosphorylates XLG2 N terminus in vitro. XLG2$^{1-203}$-HIS was incubated with HIS-BIK1 and HIS-BIK1$^{K105E}$ in the presence of $^{32}$P-γ-ATP and analyzed by autoradiography. CBB, coomassie brilliant blue. (D) BIK1 phosphorylates XLG2 at Ser148 and Ser150 in vitro. XLG2$^{1-203}$-HIS was incubated with HIS-BIK1 and HIS-BIK1$^{K105E}$ in kinase reaction buffer. Protein phosphorylation was detected by anti-pSer148 and pSer150 immunoblots. (E) XLG2 phosphorylation is required for flg22-induced ROS. xlg2 mutant plants were transformed with WT (NP::XLG2-L34), non-phosphorylatable (4A-L1 and 4A-L7), or phospho-mimicking (4D-L7 and 4D-L9) forms of XLG2 under control of the native XLG2 promoter. Independent T2 lines were examined for flg22-induced ROS burst and peak relative luminescence unit (RLU) values are shown. (mean ± SD; n ≥ 6; p<0.05, Student's t-test; different letters indicate significant difference). (F) XLG2 phosphorylation is required for Pst resistance. xlg2/3 double mutant plants were transformed with WT (XLG2-L3) or non-phosphorylatable (4A-L6 and 4A-L7) form of XLG2 under control of the native XLG2 promoter. Independent T2 lines were inoculated with Pst, and bacterial populations in leaves were measured 3 days post inoculation. (mean ± SD; n ≥ 6; p<0.05, Student's t-test; different letters indicate significant difference). (G) XLG2 interacts with RbohD in Arabidopsis plants. rbohD plants were transformed with the FLAG-RbohD transgene under control of the RbohD native promoter. The resulting plants were used for Co-IP assay. Each experiment was repeated two (C, G) or three (A, B, D–F) times, and data of one representative experiment are shown.

The following source data and figure supplements are available for figure 5:

**Source data 1.** Raw data and exact p value of *Figure 5E* and *F*.

*Figure 5 continued on next page*

*Figure 5 continued*

**Figure supplement 1.** The N terminus of XLG3, but not XLG1, is phophorylated upon flg22-treatment.
**Figure supplement 2.** Phospho-sites in XLG2 isolated from flg22-treated protoplasts.
**Figure supplement 3.** Mutations that block or mimic XLG2 phosphorylation do not impact BIK1 stability and XLG2-BIK1 interaction.
**Figure supplement 4.** XLG2/3 interact with RbohD in Nb plants.

introduced WT and the non-phosphorylatable *XLG2-FLAG* transgene into *xlg2/3* double mutants, and inoculated independent lines with *Pst*. While the line carrying the WT *XLG2-FLAG* transgene was fully restored in resistance to *Pst* that was indistinguishable from the WT non-transgenic plants, the two lines transformed with the non-phosphorylatable *XLG-FLAG* transgene were only marginally more resistant to *Pst* (*Figure 5F*). These results indicated that the phosphorylation is required, but not sufficient, for full function of XLG2. We next asked whether the phospho-site mutants affect BIK1 stability or XLG2-BIK1 interaction. When expressed in protoplasts isolated from *xlg2 xlg3* plants, the phospho-site mutants allowed similar BIK1-HA accumulation compared to protoplasts expressing the WT XLG2 protein (*Figure 5—figure supplement 3*). These mutants also showed normal interaction with BIK1 in the absence of flg22 and normal dissociation from BIK1 in the presence of flg22. The results suggest that the phosphorylation of these sites affected neither BIK1 stability nor interaction with BIK1. Because XLG2 dissociates from BIK1 following activation by flg22 (*Figure 2B*), we reasoned that XLG2 may have additional effectors. Indeed, XLG2 interacted strongly with RbohD in Arabidopsis plants (*Figure 5G*) and *Nb* plants (*Figure 5—figure supplement 4*), providing an explanation that XLG2 phosphorylation regulates ROS burst independent of BIK1 stability. The XLG2-RbohD interaction was detected in the absence of flg22 treatment, indicating that the XLG2 constitutively interacts with RbohD.

## Discussion

In this study we show that the Arabidopsis heterotrimeric G proteins positively regulate plant immunity by directly interacting with the FLS2-BIK1 immune receptor complex. The heterotrimeric G proteins regulate FLS2-mediated immunity through at least two mechanisms. Before activation, XLG2/3, AGB1, and AGG1/2 positively regulate BIK1 stability by attenuating proteasome-dependent degradation of BIK1. Upon activation of the FLS2-BIK1 complex by flg22, BIK1 directly phosphorylates XLG2 in the N terminus to regulate flg22-induced ROS production, likely through the XLG2-RbohD interaction (*Figure 6*).

### Coupling of heterotrimeric G proteins to FLS2

Heterotrimeric G proteins play important role in signal transduction in plants and animals. Paradoxically, plants do not possess functional GPCRs. The maize Gα protein CT2 interacts with the receptor kinase CLV1 to control shoot meristem development (*Bommert et al., 2013*). GPA1 and AGG1/2 have been shown to interact with BAK1 and CERK1 in yeast two-hybrid and Bi-Fluorescence Complementation assays, although the biological significance of this remains unknown (*Aranda-Sicilia et al., 2015*). We show that XLG2 directly interacts with FLS2 and BIK1 to regulate FLS2-mediated immunity. Thus, these studies collectively support that plant receptor kinases fulfil the roles of GPCRs by directly coupling to heterotrimeric G proteins.

Importantly, we show that the G proteins coupled to the FLS2 receptor complex are subject to multiple regulations during FLS2 signaling. The interaction between XLG2/3 and AGB1, which leads to the formation of XLG2/3-AGB1-AGG1/2 heterotrimers (*Maruta et al., 2015*; *Chakravorty et al., 2015*), is dynamically regulated by flg22. The flg22-induced dissociation of XLG2 from AGB1 likely reflects an increase of GTP-bound form of Gα which dissociates from the Gβγ dimer upon receptor activation. Secondly, the interactions of the G proteins with FLS2 and BIK1 are also dynamically regulated by flg22. The flg22-induced dissociation of the G proteins from BIK1 coincides with the flg22-induced BIK1 phosphorylation, suggesting that the flg22-induced BIK1 phosphorylation triggers the

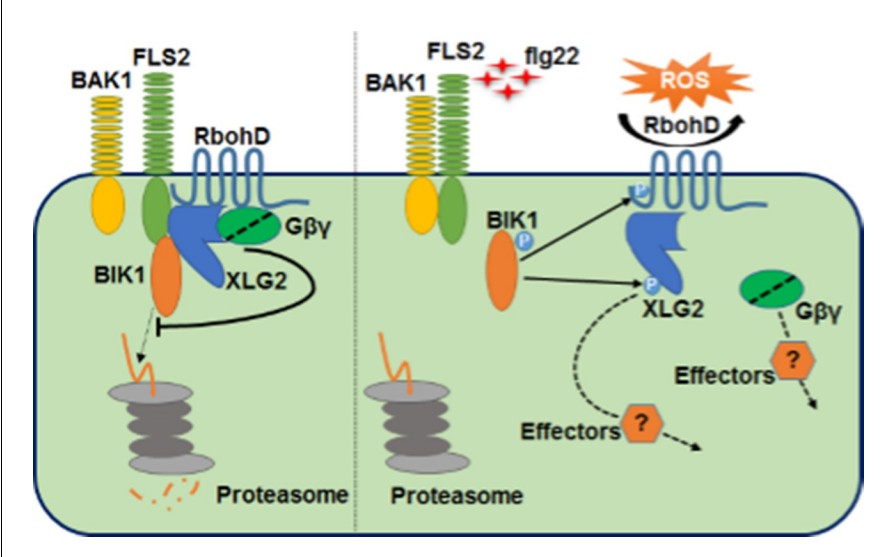

**Figure 6.** Model for G protein-coupled FLS2 signaling. In the pre-activation state, the heterotrimeric G proteins composed of XLG2/3, AGB1, and AGG1/2 interact with the FLS2-BIK1 complex. Stimulation by flg22 induces BAK1-FLS2 interaction and activation of the receptor complex. This leads to the activation of the G proteins and phosphorylation of XLG2 in the N terminus. The activated G proteins dissociate from the receptor complex and regulate RbohD and other downstream effectors to positively modulate immune responses.

dissociation of the G proteins. Indeed, de-phosphorylation of BIK1 strongly enhanced its interactions with both XLG2 and AGB1 in vitro. The decoupling of the G proteins from the receptor maybe necessary for the regulation of down-stream effectors.

## Heterotrimeric G proteins stabilizes BIK1 in the pre-activation state

BIK1 is subject to proteasome-dependent degradation (*Monaghan et al., 2014*). Our genetic analyses indicated that XLG2, AGB1 and AGG1/2 are required for BIK1 stability in the pre-activation state. Loss of *XLG2* and *AGB1* leads to accelerated degradation of BIK1, and this is blocked by the addition of proteasome inhibitors, indicating that the G proteins attenuating the proteasome-dependent degradation of BIK1. FLS2 is also subject to proteasome-dependent degradation (*Lu et al., 2011*). However, FLS2 accumulation is not affected by the *agb1* mutation, indicating that the G proteins do not generally regulate proteasome-mediated protein degradation. These observations raise an interesting question as to whether the G proteins specifically impede the ubiquitination of BIK1 or loading of BIK1 to proteasome. Importantly, introduction of a transgenic copy of BIK1 largely restored flg22-triggered immune responses and disease resistance in *xlg2* and *agb1* mutants, demonstrating that a major function of the heterotrimeric G protein is to maintain the signaling competence of the FLS2-BIK1 complex in the pre-activation state.

## Flg22-induced phosphorylation in XLG2 is required for optimum ROS production and *Pst* resistance

In addition to the ligand-induced dissociation of XLG2 from AGB1, flg22 induces the phosphorylation of XLG2 in the N terminus. This phosphorylation is likely caused by the BIK1 family kinases, as two of the major phosphor-sites induced by flg22 are also phosphorylated by BIK1 in vitro. Non-phosphorylatable XLG2 mutants are compromised in flg22-induced ROS production and disease resistance to *Pst*, indicating that the phosphorylation plays a positive role. However, mutations in the phospho-sites did not affect XLG2-BIK1 interaction nor BIK1 stability, suggesting that the phosphorylated XLG2 regulates components other than BIK1. Indeed, XLG2 constitutively interacts with RbohD, suggesting that the phosphorylated XLG2 further enhances ROS production by modulating RbohD. RbohD is known to be subject to multiple regulations including phosphorylation by CPK5 and BIK1 (*Dubiella et al., 2013*; *Kadota et al., 2014*; *Li et al., 2014*) and calcium binding to EF-

hand (*Ogasawara et al., 2008*). The results presented here show that RbohD is additionally regulated by a phosphorylated XLG2, although the underlying mechanism remains unknown.

## Conclusion

The Arabidopsis heterotrimeric G proteins reported here are analogous to the animal counterpart in that, upon activation by receptors, dissociate from each other and activate downstream effectors. Unlike the animal heterotrimeric G proteins, however, the Arabidopsis heterotrimeric G proteins additionally regulate signaling competence of the FLS2-BIK1 complex prior to receptor activation. Taken together, our results highlight remarkable similarities and striking differences in heterotrimeric G protein-coupled receptor signaling in animals and plants as a result of independent evolution.

# Materials and methods

## Plant materials

Arabidopsis plants used in this study include WT (Col-0), *xlg2*, *xlg2 xlg3* (*Ding et al., 2008*), *xlg3* (SALK_107656c), *agb1-2* (*Ullah et al., 2003*), *gpa1-3* (*Jones et al., 2003*) and *agg1 agg2* mutants (*Trusov et al., 2007*), and *NP::BIK1-HA* (*Zhang et al., 2010*) and *NP::FLAG-RbohD* transgenic lines (*Li et al., 2014*).

Nicotiana benthamiana plants used for luciferase-complementation assay and Arabidopsis plants used for ROS burst, protoplast preparation and bacterial infection assays were grown in soil at 23°C and 70% relative humidity with 10/14 hr day/night photoperiod for 4–5 weeks. Arabidopsis seedlings used for BIK1 stability and in vitroprotein degradation assay were grown in half Murashige-Skoog (MS) plates at 23°C with 16/8 hr day/night photoperiod for 7–10 days.

## Constructs and transgenic plants

To generate constructs for transient expression in protoplast, coding sequences of desired genes are amplified by PCR and cloned into the pUC19-35S-FLAG/HA-RBS vector. For GST and HIS fusion constructs, coding sequences were PCR-amplified and cloned into pGex 6P-1 (for GST fusion constructs) or pET28a (for HIS fusion constructs).

For complementation of *xlg2* (related with *Figure 1—figure supplement 1*), *XLG2* genomic sequence containing 2 kb native promoter and 0.5 kb 3'UTR was cloned into pENTY vector using pENTY Directional Cloning Kit (Invitrigen.Carlsbad, CA), transferred to pFAST-G01 vector using Gateway LR Clonase (Invitrogen) and introduced into *xlg2* mutant plants by *Agrobacterium*-mediated transformation.. To generate *XLG2* transgenic plants containing mutated phosphosites (*Figures 5E and F*), a fragment containing 2 kb native promoter and coding sequence of *XLG2* was fused to FLAG tag at the C terminus followed by the RBS terminator and cloned into pENTY vector. Mutations were introduced into pENTY-XLG2-FLAG by site-directed mutagenesis. All forms of *XLG2-FLAG* were transferred to pFAST-G01 and introduced into *xlg2* or *xlg2 xlg3* mutant plants by *Agrobacterium*-mediated transformation.

To generate *BIK1-HA* transgenic plants in *agb1, gpa1-3* and *agg1 agg2* background, the *NP:: BIK1-HA* transgene (*Zhang et al., 2010*) was introduced into *agb1, agg1/2 and gpa1-3* by crossing. Homozygous mutants and WT plants containing the transgene in the F3 generation were used for BIK1 stability and immune response assays. To express *BIK1-HA* under the 35S promoter, the cDNA of *BIK1* was amplified by PCR and cloned into pCambia1300 with a 3x HA tag. Transgenic plants expressing 35S::*BIK1-HA* were generated by transforming WT or *agb1-2* mutant plants with the pCambia1300-BIK1-HA construct. Transgenic lines in wild type and *agb1-2* background with similar expression levels of the *BIK1-HA* transgene were identified by quantitative RT-PCR analysis.

To generate *BIK1-HA* transgenic plants in *xlg2 xlg3* background, the pCAMBIA1300-NP::BIK1-HA-RBS (*Zhang et al., 2010*) construct was introduced into WT and *xlg2 xlg3* plants by *Agrobacterium*-mediated transformation. Independent lines in T2 generation were used for ROS assay and flg22-protection assay.

## Oxidative burst assay

Leaf strips form 4- to 5-week-old soil-grown *Arabidopsis* plants were incubated in water for 12 hr in 96-well plate before treated with luminescence detection buffer (20 µM luminol and 10 mg/ml

horseradish peroxidase) containing 1 μM flg22 as described (*Zhang et al., 2007*). Luminescence was recorded by using GLOMAX 96 microplate luminometer (Promega, Madison, WI).

## Flg22-protection assay

4- to 5-week-old plants were pre-treated with water or 1 μM flg22 1 day before infiltration with *P. syringae* DC3000 at a concentration of $1\times10^6$/ml, and bacterial population was determined 2 days after bacterial inoculation.

## Plant protein extraction and co-immunoprecipitation assay

Total protein was extracted from ground transgenic Arabidopsis seedlings or protoplasts with protein extraction buffer (50 mM HEPES [pH 7.5], 150 mM KCl, 1 mM EDTA, 0.5% Trition-X 100, 1 mM DTT, proteinase inhibitor cocktail). Supernatants were collected by centrifugation at 13,000 rpm for 15 min. Total protein was separated in SDS-PAGE gel and detected by immunoblot by using the indicated antibodies.

For immunoprecipitation in protoplasts, Arabidopsis protoplasts were transfected with 50–100 μg indicated plasmids, incubated overnight, and then treated with water or 1 μM flg22 for 10 min. Total protein was incubated with agarose-conjugated anti-FLAG antibody (Sigma, St. Louis, MO) for 4 hr, washed 6 times with extraction buffer and eluted with 3× FLAG peptide (Sigma) for 30 min. The immunoprecipitates were separated in a SDS-PAGE gel and detected with immunoblot.

## Mass spectrometric analyses for XLG2-interacting proteins and phosphosite detection

XLG2-FLAG plasmid was transfected into WT Arabidopsis protoplasts and incubated at room temperature for 14 hr. Total protein was extracted with IP buffer I (50 mM HEPES [pH 7.5], 50 mM NaCl, 10 mM EDTA, 0.2% Triton X-100, 0.1 mg/mL Dextran (Sigma), proteinase inhibitor cocktail) and the immunoprecipitation was performed as previously reported (*Liu et al., 2009*). To identify XLG2 phosphosites, protoplasts were treated with 1 μM flg22 for 10 min before extraction. Total protein was incubated with 70 μl agarose-conjugated anti-FLAG antibody for 4 hr, washed with IP buffer II (50 mM HEPES [pH 7.5], 50 mM NaCl, 10 mM EDTA, 0.1% Triton X-100, proteinase inhibitor cocktail) for 2 times, IP buffer III (50 mM HEPES [pH 7.5], 150 mM NaCl, 10 mM EDTA, 0.1% Triton X-100, proteinase inhibitor cocktail) for 2 times and eluted with 3× FLAG peptide for 45 min. The immunoprecipitates were separated in 10% NuPAGE gel (invitrogen) and subject to Mass Spectrometric analysis as previously described (*Li et al., 2014*).

## GST pull-down and in vitro phosphorylation assays

For GST pull-down assay, GST-BIK1, GST-BIK1$^{K105E}$, GST-FLS2KD and XLG2CT-HIS proteins were purified using the glutathione agarose beads and Ni-NTA agarose beads. 10 μg GST- and HIS-tagged proteins were incubated with 30 μl glutathione agarose beads in GST buffer (25 mM Tris-HCl, 100 mM NaCl, 1 mM DTT, pH 7.5) for 3 hr. The beads were washed 5–6 times with GST buffer and eluted with elution buffer (25 mM Tris-HCl, 100 mM NaCl, 1 mM DTT, 15 mM GSH, pH 7.5). Samples were separated in SDS-PAGE gel and detected by anti-HIS immunoblot.

For in vitro phosphorylation assay, 200 ng HIS-BIK1 or HIS-BIK1$^{K105E}$ was incubated with 2 μg XLG2$^{1-203}$-HIS protein in the kinase reaction buffer (25 mM Tris-HCl, 10 mM MgCl$_2$, 1 mM DTT, 100 mM ATP, pH 7.5) for 30 min. The phosphorylation of XLG2$^{1-203}$ was detected by autoradiograph (by adding γ-P$^{32}$ ATP in kinase reaction buffer) or by immunoblot using phosphosite-specific antibodies.

## Luciferase complementation assay

The coding sequences of indicated genes are cloned into pCAMBIA1300-35S-Cluc-RBS or pCAMBIA1300-35S-HA-Nluc-RBS and are introduced into *Agrobacterium tumefaciens* strain GV3101. The assay was carried out as previously described (*Chen et al., 2008*), Agrobacterial strains carrying the indicated constructs were infiltrated into *Nb* leaves. Leaf discs were taken 2 days later, incubated with 1 mM luciferin in a 96-well plate for 5–10 min, and luminescence was recorded with the GLOMAX 96 microplate luminometer.

## Phosphosite antibodies

XLG2 phosphosite-specific antibodies were produced by Abmart (China) as previous described (*Li et al., 2014*) using the following peptides:

|  | Immunization peptide | Control peptide |
| --- | --- | --- |
| p148 | -ADFRL(**pS**)PSSPL | -ADFRLSPSSPL |
| p150 | -FRLSP(**pS**)SPLSA | -FRLSPSSPLSA |
| p151 | -RLSPS(**pS**)PLSAS | -RLSPSSPLSAS |

## In vitro protein degradation assay

In vitro protein degradation assay was performed as previous described (*Wang et al., 2009*) with slightly modification. 1-week-old seedlings of indicated genotype were ground in 200–300 µl extraction buffer (25 mM Tris-HCl [pH 7.5], 10 mM NaCl, 10 mM MgCl$_2$, 4 mM PMSF, 5 mM DTT, and 10 mM ATP). Supernatants were collected by centrifugation at 14,000 rpm for 10 min and total protein concentration was determined by Bio-Rad protein assay and adjusted to a final concentration of 1µg protein/µl. 300 ng recombinant HIS-BIK1 or PBL20-HIS protein was incubated with 100 µl total extract at 22°C, and equal amounts of samples were withdrawn at the indicated times for anti-HIS immunoblot analysis.

## RNA isolation and qRT-PCR

4-week-old WT Arabidopsis plants were treated with flg22 for 0 hr and 3 hr and total RNA was extracted by using RNeasy Plant Mini Kit (Qiagen, Germany). First-strand cDNA synthesis was performed using SuperScript III RNA transcriptase (Invitrogen) following manufacturer's instructions. Real-Time PCR was performed using and specific primers and SYBR Premix Ex Taq Kit (TaKaRa, Japan).

## Oligonucleotide primers

Primers for Real-Time PCR:
ACT8-RT-F: 5'-TGTGACAATGGTACTGGAATGG-3'
ACT8-RT-R: 5'-TTGGATTGTGCTTCATCACC -3'
XLG1-RT-F: 5'-TGATGGTGAGGATTGTGAATTGA-3'
XLG1-RT-R: 5'-TTCCCAATCCGGTACTAACGG-3'
XLG2-RT-F: 5'-ATTGCTAATGTGCCACGAGCT-3'
XLG2-RT-R: 5'-ACGAGAGGTGCCACTGGGTAA-3'
XLG3-RT-F: 5'-CCGGTTGTGAAATTCAAACCTG-3'
XLG3-RT-R: 5'-TCCCTCTCTGTCTCTGCCTCC-3'
BIK1-HA-RT-F: 5'-CAGGACAACTTGGGAAAACCG-3'
BIK1-HA-RT-R: 5'-TAGGATCCTGCATAGTCCGGG-3'
Primers for site-directed mutagenesis
XLG2 (S141A)-F:
5'-ATGTACCAGAAGAAGTGAAAGCTCCTGCTGATTTTCGGTTATC-3'
XLG2 (S141A)-R:
5'-GATAACCGAAAATCAGCAGGAGCTTTCACTTCTTCTGGTACAT-3'
XLG2 (S148A)-F:
5'-TCCTGCTGATTTTCGGTTAGCACCATCATCACCATTGTCTG-3'
XLG2 (S148A)-R:
5'-CAGACAATGGTGATGATGGTGCTAACCGAAAATCAGCAGGA-3'
XLG2 (S150A/S151A)-F:
5'-GCTGATTTTCGGTTATCACCAGCAGCACCATTGTCTGCATCAGCGAGA-3'
XLG2 (S150A/S151A)-R:
5'-TCTCGCTGATGCAGACAATGGTGCTGCTGGTGATAACCGAAAATCAGC-3'
XLG2-3A (S148A/S150A/S151A)-F:

5'-AAAGTCCTGCTGATTTTCGGTTAGCACCAGCAGCACCATTGTCTGCATCAGCGAGA
GA-3'
XLG2-3A (S148A/S150A/S151A)-R:
5'-TCTCTCGCTGATGCAGACAATGGTGCTGCTGGTGCTAACCGAAAATCAGCAGGACT
TT-3'
XLG2-3D (S148D/S150D/S151D)-F:
5'-AAAGTCCTGCTGATTTTCGGTTAGACCCAGACGACCCATTGTCTGCATCAGCGAGA
GA-3'
XLG2-3D (S148D/S150D/S151D)-R:
5'-TCTCTCGCTGATGCAGACAATGGGTCGTCTGGGTCTAACCGAAAATCAGCAGGACT
TT–3'
XLG2 (S141D)-F:
5'-ATGTACCAGAAGAAGTGAAAGATCCTGCTGATTTTCGGTTATC-3'
XLG2 (S141D)-F:
5'-GATAACCGAAAATCAGCAGGATCTTTCACTTCTTCTGGTACAT-3'

## Acknowledgements

We wish to thank Yiji Xia for providing *xlg2* and *xlg3* mutants. Yuli Ding and Yufei Yu for technical assistance. JMZ was funded by the Natural Science Foundation China (Grant No. 31320103909), Chinese Ministry of Science and Technology (Grant No. 2015CB910201), Chinese Academy of Sciences (Strategic Priority Research Program Grant No. XDB11020200 and Joint Scientific Thematic Research Programme Grant No GJHZ1311), and State Key Laboratory of Plant Genomics. YZ was funded by Natural Sciences and Engineering Research Council (Grant No. 249922).

## Additional information

### Funding

| Funder | Author |
| --- | --- |
| National Natural Science Foundation of China | Jian-Min Zhou |
| Natural Sciences and Engineering Research Council of Canada | Yuelin Zhang |
| Chinese Academy of Sciences | Jian-Min Zhou |
| Ministry of Science and Technology of the People's Republic of China | Jian-Min Zhou |

The funders had no role in study design, data collection and interpretation, or the decision to submit the work for publication.

### Author contributions

XL, Designed and performed the majority of the experiments, Acquisition of data, Analysis and interpretation of data, Drafting or revising the article; PD, Designed and performed the majority of the experiments, Acquisition of data, Analysis and interpretation of data; KL, Contributed to the analyses of BIK1 stability in plants, Acquisition of data, Analysis and interpretation of data; JW, Performed in vitro protein degradation assays, Acquisition of data, Analysis and interpretation of data; MM, LLi, Performed part of the protein-protein interaction studies, Acquisition of data, Analysis and interpretation of data; LLi, Performed mass-spectrum analyses, Acquisition of data, Analysis and interpretation of data; ML, XZ, Participated in the characterization of xlg2 mutant, Acquisition of data, Analysis and interpretation of data; SC, Performed mass-spectrum analyses, Conception and design, Analysis and interpretation of data, Drafting or revising the article; YZ, J-MZ, Coordinated the research and wrote the paper, Conception and design, Analysis and interpretation of data

**Author ORCIDs**

Pingtao Ding, http://orcid.org/0000-0002-3535-6053

Jian-Min Zhou, http://orcid.org/0000-0002-9943-2975

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
