## [Decision Letter]

Thank you for submitting your work entitled "Arabidopsis heterotrimeric G proteins directly regulate FLS2-mediated immunity" for consideration by *eLife*. Your article has been favorably evaluated by Ian Baldwin as the Senior editor and three reviewers, one of whom, Thorsten Nürnberger, is a member of our Board of Reviewing Editors, and another is Roger Innes.

The reviewers have discussed the reviews with one another and the Reviewing Editor has drafted this decision to help you prepare a revised submission.

In general, the referees feel that your MS reports important novel insight into the role of G-protein signaling (XLG2, AGB1, AGG1/2) in plants in general and in immune signaling in particular. The role of G-proteins in regulating BIK1 function is novel and exciting. Particularly strong are the data supporting a role of G-proteins in regulating pre-stimulation processes in immunity. Compared to this, however, a role of G-proteins in signaling post flg22 treatment appears rather vague and mostly correlative. For example, what is the evidence that phosphorylated XLG2 (pXLG2) is required for interaction with RbohD? The latter is important as evidence for post-activation function of G-proteins is mainly correlative (flg22 treatment results in phosphorylation of the N-terminus of XLG2 and XLG2 interacts with RbohD apparently even in the absence of flg22). Likewise, how does XLG2 binding to RbohD affect RbohD activity and why don't the phosphomimic forms of XLG2 show high basal levels of ROS production in the absence of flg22? Do XLG2 phosphomimic plants show phenotypic alterations or changes in pathogen resistance?

There are also concerns regarding the novelty of some of the findings presented when comparing those to published work.

1) Are there literature data available on how *xlg1, agg3*, and *gpa1* mutants perform in the specific assays reported here in Figure 1? If not, experiments shown in Figure 1 should be conducted on these genotypes. If yes, these data should be referred to. Likewise, the results of Figure 1 were already reported by Maruta et al., 2015 in their Figure 1 and by the senior author in Figure 3 of Liu et al. The results of Figure 1 were already reported by Maruta et al., 2015 in their Figure 2 as well as by the senior author in Figure 3 of Liu et al. Interaction between XLG2 and AGB1 in Figure 1 was already reported (using a different protein-protein interaction system) by Maruta et al. 2015 and by Zhu et al., 2009. Therefore, none of these results are novel. The reader would not know this, however, from the way the manuscript is written. For example, the authors state that the results of Figure 1 "are consistent with the reported role of XLG2 in flg22-induced immune responses" (Maruta et al., 2015), rather than correctly stating that what they have done is to replicate Maruta's already-published results.

2) There are also concerns that the interpretation of Figure 1 is erroneous. The authors do not see a phosphorylation band in the *agb1* lanes and conclude from this that AGB1 is needed for flg22-dependent RbohD phosphorylation. However, if one looks at the basal amount of FLAG-RhohD present in the *agb1* plants, it is less than the amount present in the Col-0 background. I suspect that if they enhanced their blots so that the signal intensity for FLAG-RhohD in *agb1* was as great as in Col-0 then they would see phosphorylation of pS39 in the *agb1* background, i.e. that AGB1 is not, in fact, required for flg22-dependent RbohD phosphorylation. In fact, if one looks closely, there is even in the present figure a hint of flg22-dependent pS39 phosphorylation in the *agb1* lane.

3) Zhu et al. 2009 reported that the *xlg2* loss-of-function mutation enhances susceptibility to *P. syringae*. This is a very important finding that sets the stage for all the present work in this manuscript and this finding should be explicitly described and acknowledged in the Introduction. The authors are similarly vague about the important contributions of Torres and Dangl and colleagues (MPMI 2013) with regard to AGB1 interaction with NADPH oxidases and effects of *agb1* mutation on ROS production. Those results should also be explicitly described in the Introduction.

---

## [Author Response]

*In general, the referees feel that your MS reports important novel insight into the role of G-protein signaling (XLG2, AGB1, AGG1/2) in plants in general and in immune signaling in particular. The role of G-proteins in regulating BIK1 function is novel and exciting. Particularly strong are the data supporting a role of G-proteins in regulating pre-stimulation processes in immunity. Compared to this, however, a role of G-proteins in signaling post flg22 treatment appears rather vague and mostly correlative. For example, what is the evidence that phosphorylated XLG2 (pXLG2) is required for interaction with RbohD? The latter is important as evidence for post-activation function of G-proteins is mainly correlative (flg22 treatment results in phosphorylation of the N-terminus of XLG2 and XLG2 interacts with RbohD apparently even in the absence of flg22). Likewise, how does XLG2 binding to RbohD affect RbohD activity and why don't the phosphomimic forms of XLG2 show high basal levels of ROS production in the absence of flg22? Do XLG2 phosphomimic plants show phenotypic alterations or changes in pathogen resistance?* XLG2 interacts constitutively with RbohD, as shown in Figure 5 and Figure 5—figure supplement 5. So a phosphorylated XLG2 may alter RbohD activity through a phosphorylation-induced conformational change or by recruiting other components. At this point, we do not know the real answer and will have to keep it vague. Nonetheless, we have clarified our results to state that the interaction is constitutive (at the end of the Results section and in the subsection “Flg22-induced phosphorylation in XLG2 is required for optimum ROS production and Pst resistance”) and that the mechanism by which the phosphorylated XLG2 regulate RbohD activity remains unknown (in the aforementioned subsection). Phospho-mimicking form of XLG2 showed identical phenotypes as did WT plants. This is likely because XLG2 effectors, such as RbohD, are regulated by multiple signal input. The phosphorylation of XLG2, while necessary, is not sufficient for the activation of RbohD. This point has been clearly stated in the text (subsections “XLG2 phosphorylation is required for disease resistance to Pst and optimum ROS production upon flg22 induction “and “Flg22-induced phosphorylation in XLG2 is required for optimum ROS production and Pst resistance“).

*There are also concerns regarding the novelty of some of the findings presented when comparing those to published work.*

*1) Are there literature data available on how xlg1, agg3, and gpa1 mutants perform in the specific assays reported here in Figure 1? If not, experiments shown in Figure 1 should be conducted on these genotypes. If yes, these data should be referred to. Likewise, the results of Figure 1 were already reported by Maruta et al., 2015 in their Figure 1 and by the senior author in Figure 3 of Liu et al. The results of Figure 1 were already reported by Maruta et al., 2015 in their Figure 2 as well as by the senior author in Figure 3 of Liu et al. Interaction between XLG2 and AGB1 in Figure 1 was already reported (using a different protein-protein interaction system) by Maruta et al. 2015 and by Zhu et al., 2009. Therefore, none of these results are novel. The reader would not know this, however, from the way the manuscript is written. For example, the authors state l. 88-90 that the results of Figure 1 "are consistent with the reported role of XLG2 in flg22-induced immune responses" (Maruta et al., 2015), rather than correctly stating that what they have done is to replicate Maruta's already-published results.*

Literature data on *xlg1, gpa1* and *agg3* are now described and cited (Introduction, third paragraph, and Results, first paragraph).

*2) There are also concerns that the interpretation of Figure 1 is erroneous. The authors do not see a phosphorylation band in the agb1 lanes and conclude from this that AGB1 is needed for flg22-dependent RbohD phosphorylation. However, if one looks at the basal amount of FLAG-RhohD present in the agb1 plants, it is less than the amount present in the Col-0 background. I suspect that if they enhanced their blots so that the signal intensity for FLAG-RhohD in agb1 was as great as in Col-0 then they would see phosphorylation of pS39 in the agb1 background, i.e. that AGB1 is not, in fact, required for flg22-dependent RbohD phosphorylation. In fact, if one looks closely, there is even in the present figure a hint of flg22-dependent pS39 phosphorylation in the agb1 lane.*

We have quantified the level of phosphorylation for each sample, normalized to the amounts of total FLAG-RbohD protein, and the numbers are now provided in Figure 1.

3) Zhu et al. 2009 reported that the xlg2 loss-of-function mutation enhances susceptibility to P. syringae. This is a very important finding that sets the stage for all the present work in this manuscript and this finding should be explicitly described and acknowledged in the Introduction. The authors are similarly vague about the important contributions of Torres and Dangl and colleagues (MPMI 2013) with regard to AGB1 interaction with NADPH oxidases and effects of agb1 mutation on ROS production. Those results should also be explicitly described in the Introduction.

These two important studies are now explicitly described in the Introduction (third paragraph).